# Bandit Sampling For Faster neural network Training with SGD

**Francisco Calderon, Anila Joshi, Vignesh Ganapathiraman**
Amazon
Seattle, Washington, USA
`{fccal,vignesga, anilajo}@amazon.com`

## Abstract

Importance sampling is a valuable technique in deep learning that involves sampling useful training examples more frequently to improve learning algorithms. However, obtaining reliable sample importance estimates early on in training can be challenging, as existing importance sampling methods can be computationally expensive and slow to converge. In this work, we propose a novel sampling schemed based on Multi-arm bandits (MAB). The proposed sampler is able to achieve higher validation accuracies significantly earlier in the training, compared to baselines.

## 1 Introduction

Deep learning model training has become more efficient lately thanks to stochastic gradient-based optimizers like SGD and Adam, which have aided in improving learning algorithms. However, during training, uniformly sampling a mini-batch of training examples in each iteration may cause the learning algorithm to converge slowly, as not all examples contribute equally to the prediction task of the deep learning model (Ganapathiraman et al.; Katharopoulos & Fleuret, 2018; Alain et al., 2015). To address this issue, importance sampling techniques have been proposed, which involves sampling useful training examples more frequently based on their importance score. Importance sampling methods are particularly useful when training on a limited budget, such as during hyperparameter optimization or model selection.

One of the challenges in importance sampling for neural network training is to obtain reliable estimates of sample importance early on in training. Existing importance sampling algorithms rely on local metrics such as loss and gradient norms to estimate sample importance, which can be computationally expensive and often require several iterations to achieve significant improvements (Katharopoulos & Fleuret, 2017; 2018; Alain et al., 2015; Chang et al., 2017). This can lead to slower convergence of the learning algorithm and longer training times.

In this work, we explore the idea of using bandits to sample mini-batches for training neural networks using training loss as the guiding signal. Similar directions have been explored in the past. Liu et al. (2020b) used bandits for speeding up GNN training. Recently, Liu et al. (2020a) proposed a bandit-based sampling strategy specifically for ADAM with promising performance. In the proposed RL-based sampler, the Multi-Arm Bandit Sampler (MABSampler), loss is used as the reward to update and improve the sampler at each iteration. MABSampler has shown to consistently outperform other importance sampling techniques such as SGD and p-SGD (Chang et al., 2017), achieving maximum accuracy in significantly fewer iterations than other methods

## 2 Method

The training architecture using the proposed MABSampler is shown in Figure 2. There are 2 main components:

1. **Importance sampler.** The sampler has $b$ one-arm bandits, where $b$ is the batch size. During training, bandit $\text{MAB}_i$ samples one arm, which corresponds to one training example from a pool of $N/b$ examples, which feeds into the *Batch Sampler*. The training data is split

    randomly into chunks of size $N/b$ among the bandits, with *class stratification* to ensure that each bandit has access to the same data distribution; minimizing initial sampling bias.

2. **Loss Binarizer.** The bandits are updated using a reward function, that is based on the training loss of a batch. The Loss Binarizer module takes in a per-sample training loss and binarizes it using the following function: $r(x_a) = \begin{cases} 1.0 & \text{if } \ell(x_a) \geq m_l \\ 0 & \text{Otherwise} \end{cases}$, where $m_l$ is a moving average of the training loss and $\ell(x_a)$ is the training loss of $x_a$.

## 3 EXPERIMENTS

The bandits were updated using the Thompson sampling algorithm, which gave the best results in our experiments, compared to other methods such as $\epsilon$-greedy, softmax action selection. We also tried parametric and contextual bandits, but they were really slow to train. Figure 2 (a & b) show validation accuracies on image classification benchmarks. MABSampler achieves significantly higher performance much earlier in the training compared to baselines such as p-SGD and c-SGD.

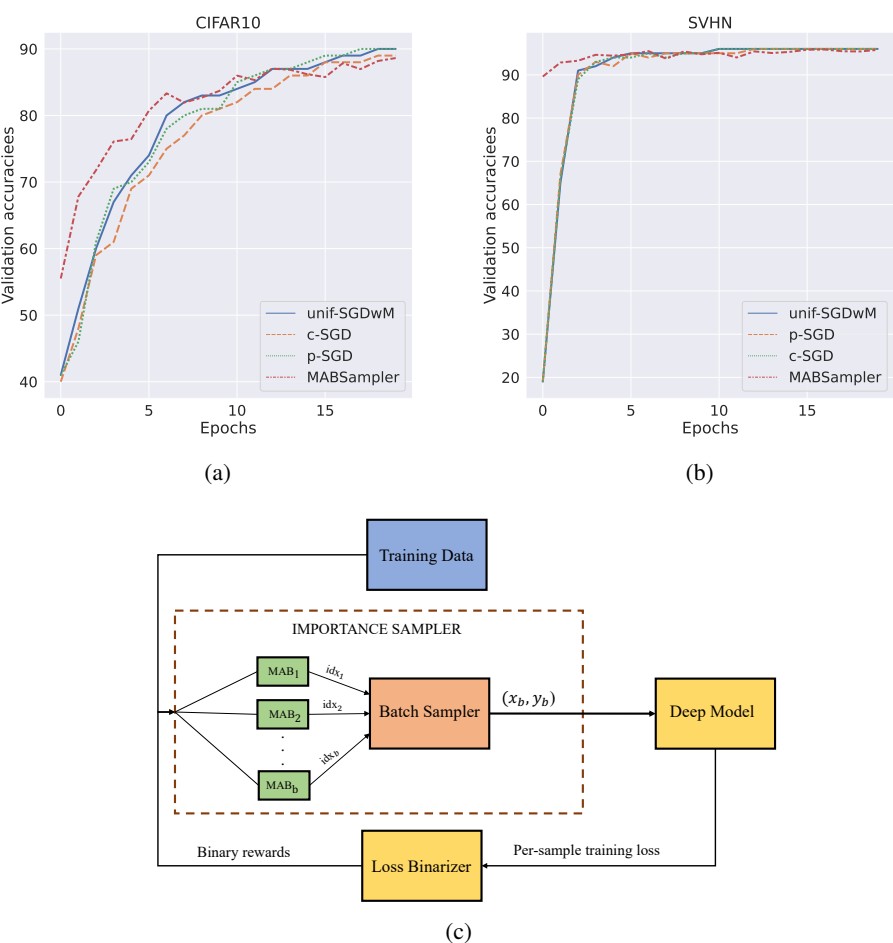

(a)                        (b)

(c)

Figure 1: (a) & (b) Validation accuracies of MAB sampler along with popular importance sampling algorithms on popular image classification benchmarks. (c) Training pipeline of the proposed Bandit sampler for SGD.

## 4 CONCLUSION

In this work, we proposed a novel bandit-based non-uniform sampling technique for training neural networks with SGD. Empirical studies show that the proposed model outperformed existing baselines significantly in earlier batches, offering great benefit for budgeted training scenarios.

URM STATEMENT

The authors acknowledge that at least one key author of this work meets the URM criteria of ICLR 2023 Tiny Papers Track.

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

## A    ADDITIONAL PLOTS FOR IMAGE CLASSIFICATION

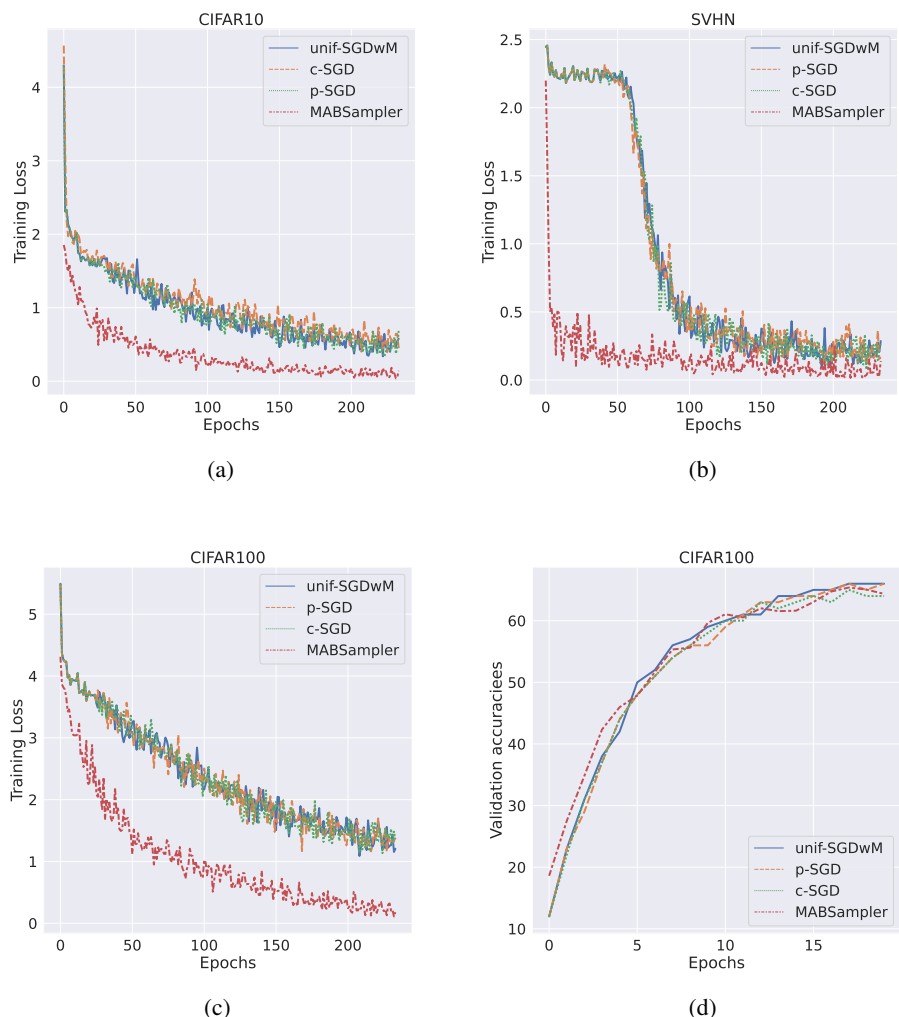

Figure 2: (a, b, c) Training loss of the proposed MAB-based sampler along with baselines on popular image classification benchmark datasets. (d). Validation accuracy on CIFAR100 dataset (additional plot from the validation accuracies plot from the main paper.)

## B    EXPERIMENT SETUP

In this section, we describe additional experimental details.

**MABSampler implementation.**    The experiments were developed using PyTorch. We implemented the MABSampler as a subclass of the `Sampler` [1] class in PyTorch. As discussed earlier in Section 3, the sampler itself made use of $b$ one arm bandits, each of which is responsible for selecting one example for a mini-batch. We took inspiration from community discussions present here `https://github.com/pytorch/pytorch/issues/7359` to implement the custom sampler in PyTorch. The individual bandits were implemented using the `Contextual Bandits` library [2]

---

[1] `https://pytorch.org/docs/stable/data.html#torch.utils.data.Sampler`
[2] `https://contextual-bandits.readthedocs.io/en/latest/`

**Baseline implementation.**    We evaluated our proposed sampler against 3 baselines. The "unif-SGDwM" baselines referes to vanilla SGD with momentum. Whereas p-SGD and c-SGD are well known analyical importance sampling techniques. We followed the implementation of `https://github.com/EricArazo/ImportanceSampling` (Arazo et al., 2021) for c-SGD and p-SGD samplers. All the samplers were trained with SGD as the base optimizer with a learning rate of $1e^{-1}$ and batch size of $128$ for all the datasets.

