# OpenReview forum: "BANDIT SAMPLING FOR FASTER NEURAL NETWORK TRAINING WITH SGD"
_ICLR.cc/2023/TinyPapers — Submitted to Tiny Papers @ ICLR 2023_

### Official Review · Reviewer_5wCv · 2023-03-27

**Confidence:** 4

**Summary Of Contributions:**

Obtaining reliable sample importance estimates early on in training can be challenging, as existing methods can be computationally expensive and slow to converge. The authors propose a novel sampling scheme based on Multi-arm bandits (MAB) that achieves higher validation accuracies significantly earlier in training compared to baselines.

**Rating:**

Clear, Correct, and Reproducible (CCR): a submission which meets the reviewing criteria

**Strengths And Weaknesses:**

*Strengths*
- The findings are clearly communicated, and the paper appropriately discusses other relevant literature.
- The ideas presented in this paper are interesting.
- The claims are justified, and the paper is well-motivated.

*Weaknesses*
- The authors do not describe the details of the experiment, making it hard to reproduce.
- Although it is minor, I wish the authors had described potential applications of MAB (e.g., hyperparameter optimization).

**Suggested Changes:**

- Please include the experiment setup to ensure that they are reproducible.
- As mentioned above, it would be great if the authors described potential applications. If we used it for hyperparameter optimization (e.g., train it for first k iterations), would the method have a better correlation against the ground-truth (e.g., full training)?

---

### Official Review · Reviewer_XDXi · 2023-04-02

**Confidence:** 3

**Summary Of Contributions:**

The authors propose a novel bandit-based approach to sampling batches to achieve early gains in accuracy, particularly useful for methods, such as HPO, that require lower fidelity approximations of final performance for cost constraints.

**Rating:**

Great Start (GS): a submission which meets some of the reviewing criteria but has room for improvement

**Strengths And Weaknesses:**

Strengths:
- The paper proposes an interesting and potentially cheap method for improving performance early-on
- The method and results have been communicated clearly

Weaknesses:
- MAB, while technically stateless RL, is a separate subfield in and of itself. The proposed approach is a MAB sampler, and calling it that would suffice. The third paragraph is misleading.
- The results are not convincing in showing that the proposed method indeed outperforms baselines. First, the performance gains early on in CIFAR datasets are suboptimal, and at least 10 measures away from the accuracy in the later epochs. I am not sure a 5th epoch approximation at 80% accuracy on CIFAR-10 would be useful to an HPO method. Additionally, the performance gains in CIFAR-100 are negligible.
- The hyperparameters of the experiments have not been communicated, making reproducibility difficult.



**Suggested Changes:**

- Please beware of \citep vs \citet. The citation style in the text is confusing
- I would suggest rephrasing the third paragraph to exclusively just talk about MAB and its use in this case
- I would recommend showing the applicability of the achieved gains on CIFAR datasets in something like HPO to demonstrate that the sampling strategy can indeed be useful

---

### Official Review · Reviewer_4heJ · 2023-04-04

**Confidence:** 3

**Summary Of Contributions:**

No Double Descent in Self-Supervised Learning  The paper investigates the Double decent hypothesis in the context of self-supervised learning and provides evidence against DD.

**Rating:**

High Potential (HP): a submission which meets the reviewing criteria and has potential to make an impact on the field

**Strengths And Weaknesses:**


Great review of the literature so far.
Limited use of architectures.


**Suggested Changes:**


Could try an architecture other than AE.

---

### Author Response · Authors · 2023-05-31
**Archival opt-in**

Thank you very much for the review and constructive feedback. We wish to opt-in for archival.

---

### Author Response · Authors · 2023-05-31
**Response to reviews**

We thank the reviewers for their efforts in reviewing this work. The suggested changes and comments helped improve our paper significantly. Below we list the major revisions made based on the feedback.

1. We updated the introduction to focus the narrative on Bandits
2. Fixed citation style issues
3. Added additional experiment details to the appendix to aid reproducibility.

There were a couple of comments suggesting detailed experiments on the benefits of the method in HPO. This is indeed a great suggestion and we plan to take this up in the future.

---

### Meta-Review · Area_Chair_dxdx · 2023-04-04

**Recommendation:** Invite to archive
**Confidence:** 3

**Metareview:**

The ideas are interesting and have a good potential. The method and experiments are mostly clearly communicated. But some reviewers are concerned that the results may be not convincing enough and the performance gains are not very significant, and some experimental details are missing.



**Summary:**

This paper proposes a bandit-based approach to sampling batches, to achieve early gains in the training.

**Reason For Not Giving A Higher Recommendation:**

Some reviewers are concerned that the results may be not convincing enough and the performance gains are not very significant, and some experimental details are missing.

**Reason For Not Giving A Lower Recommendation:**

The ideas are interesting and have a good potential. The method and experiments are mostly clearly communicated.

---

### Decision · Program_Chairs · 2023-04-08

Invite to archive